# Congenital malformations and preeclampsia associated with integrase inhibitor use in pregnancy: A single-center analysis

Christiana Smith[1]*, Angela J. Fought[2], Joyce F. Sung[2], Jennifer R. McKinney[3], Torri D. Metz[4], Kirk B. Fetters[5], Sarah Lazarus[6], Shannon Capraro[7], Emily Barr[8], Carrie Glenny[9], Jenna Buehler[2], Adriana Weinberg[1,10], for the CHIP Perinatal Medical Team[¶]

1 Department of Pediatrics, University of Colorado, Aurora, CO, United States of America, 2 University of Colorado, Aurora, CO, United States of America, 3 Baylor College of Medicine, Houston, TX, United States of America, 4 University of Utah Health, Salt Lake City, UT, United States of America, 5 Department of Medicine, Harbor-UCLA Medical Center, Torrance, CA, United States of America, 6 University of Wisconsin, Madison, WI, United States of America, 7 Department of Pediatrics, Children's Mercy Hospital, Kansas City, MO, United States of America, 8 UT Health Houston, Cizik School of Nursing, Houston, TX, United States of America, 9 Children's Hospital Colorado, Aurora, CO, United States of America, 10 Departments of Medicine and Pathology, University of Colorado, Aurora, CO, United States of America

¶ Membership of the CHIP Perinatal Medical Team is provided in the Acknowledgments.
* Christiana.Smith@childrenscolorado.org

**Data Availability Statement:** The underlying data used to reach the conclusions drawn in the manuscript derive from human research participant data. It is not ethically appropriate to

## Abstract

### Background

Antiretroviral therapy (ART) decreases perinatal HIV transmission, but concerns exist regarding maternal and infant safety. We compared the incidence of congenital malformations and other adverse outcomes in pregnancies exposed to integrase inhibitor (INSTI) versus non-INSTI ART.

### Setting

Single-site review of all pregnancies among women living with HIV between 2008 and 2018.

### Methods

We used binomial family generalized estimating equations to model the relationship of congenital anomalies and pregnancy outcomes with exposure to INSTI or dolutegravir (DTG) versus non-INSTI ART.

### Results

Among 257 pregnancies, 77 women received ≥1 INSTI (54 DTG, 14 elvitegravir, 15 raltegravir), 167 received non-INSTI, and 3 had missing data. Fifty congenital anomalies were identified in 36 infants. Infants with first-trimester DTG or any first-trimester INSTI exposure had higher odds of congenital anomalies than infants with first-trimester non-INSTI exposure (OR = 2.55; 95%CI = 1.07–6.10; OR = 2.61; 95%CI = 1.15–5.94, respectively). Infants with INSTI exposure after the second trimester had no increased odds of anomalies.

publicly share this data set because it contains potentially identifiable information from a small group of vulnerable participants from a single institution. The data set is therefore only available upon request. Investigators interested in accessing the data would need to work with the University of Colorado Office of Grants and Contracts to develop a MTA to obtain the intellectual property, and the office would need to coordinate with a member of the study team for access to the data. The Office of Grants and Contracts at the University of Colorado Anschutz Medical Campus can be contacted by phone at 303-724-0090, via email at xenia@ucdenver.edu, or by mail: Grants and Contracts, Mail Stop F428 Anschutz Medical Campus, Fitzsimons Building 13001 E 17th Place, Room W1124 Aurora, CO 80045-2571

**Funding:** Institutional funding supported this study. The CHIP clinic receives funding from award numbers: HHSN275201800001I; 5P01HD103133-02; 5U01HD068063-05; 5U01HD068040474-15S1; UM1 AI068619; ENVHL-202158734-00; X0700056; H12HA24784. Through access to University of Colorado REDCap, this project was supported by NIH/NCATS Colorado CTSA Grant Number UL1 TR002535. Its contents are the authors' sole responsibility and do not necessarily represent official NIH views. The funders had no role in study design, data collection and analysis, decision to publish, or preparation of the manuscript.

**Competing interests:** I have read the journal's policy and the authors of this manuscript have the following competing interests: Adriana Weinberg discloses research grants from GlaxoSmithKline and Merck that were paid to the University of Colorado and personal fees from Merck, Squirus and GlaxoSmithKline for work unrelated to this study. The remaining authors have declared that no competing interests exist.

Women with INSTI exposure had higher odds of preeclampsia (OR = 4.73; 95%CI = 1.70–13.19). Among women who received INSTI, grade $\geq$3 laboratory abnormalities were noted in 2.6% while receiving the INSTI and 3.9% while not receiving the INSTI, versus 16.2% in women who received non-INSTI. There was no association between INSTI exposure and other pregnancy outcomes.

## Conclusion

In our cohort, first-trimester INSTI exposure was associated with increased rates of congenital anomalies and use of INSTI during pregnancy was associated with preeclampsia. These findings underscore the need for continued monitoring of the safety of INSTI in pregnancy.

## Introduction

Antiretroviral therapy (ART) is an invaluable tool for both preventing perinatal transmission of HIV and improving maternal health, but concerns exist regarding the maternal and fetal safety of these medications. For example, protease inhibitors (PI) were associated with increased rates of preterm birth; nevirapine and efavirenz with severe hepatitis; and stavudine and didanosine with lactic acidosis [1]. Integrase strand transfer inhibitors (INSTI) are a newer class of ART that have been incompletely studied in pregnancy. The INSTI dolutegravir (DTG) quickly gained popularity for use in pregnant women due to its antiviral potency, high genetic barrier to resistance, and low side effect profile. In 2018, a birth outcomes surveillance study in Botswana reported increased rates of neural tube defects among infants exposed to DTG at the time of conception, leading some public health institutions to advise against its use in women contemplating pregnancy [2]. However, this study subsequently reported that the risk of neural tube defects was lower than initially thought, and an increased risk has not been confirmed in other large observational or retrospective studies [3–6]. DTG is now recommended as a preferred drug during pregnancy [1, 7].

Although a definitive link between DTG and neural tube defects has not been established, there has been recent interest in a potential association between DTG and other types of congenital anomalies. Reports published to date have had insufficient power to identify increases in the rate of anomalies among DTG-exposed infants over the background prevalence. Post-marketing surveillance data identified musculoskeletal, cardiac, and neurologic problems in infants exposed to DTG, but these reports are difficult to interpret given that the total number of infants exposed to these medications is unknown, so the proportion of infants with defects cannot be calculated [8]. Other observational studies have reported rates of congenital anomalies ranging from 4–24% in DTG-exposed infants [4, 9]. The SMARTT study identified a non-significant trend toward increased neurologic diagnoses, including microcephaly and strabismus, among children with in utero DTG exposure compared with other types of ART, with a greater magnitude of association for exposure during the first trimester and at conception [10].

Fewer data exist for INSTIs other than DTG, including raltegravir (RAL), elvitegravir (EVG), and bictegravir. These INSTIs have not been associated with an increased risk of congenital anomalies among the few cases reported to the Antiretroviral Pregnancy Registry or the Canadian Perinatal HIV Surveillance Programme; however, these databases are subject to reporting biases [11, 12]. Observational cohorts from the US, UK, and Ireland reported low rates (0–2.5%) of congenital anomalies among RAL- and EVG-exposed infants [13, 14]. The

French Perinatal Cohort found a concerning 2-fold higher rate of birth defects in infants exposed to RAL at conception compared to matched controls (6.4% vs. 2.3%), albeit this difference did not reach statistical significance (p = 0.08) [15]. These studies suggest that ongoing surveillance of infants with in utero INSTI exposure is needed.

Pregnancy outcomes other than congenital anomalies have also been inadequately described among women exposed to DTG or other INSTIs. Several studies reported no increased rates of stillbirth, spontaneous abortion, preterm birth, or SGA among INSTI-exposed infants [4, 5, 15–17]. In contrast, the SMARTT study identified a 2-fold higher rate of preterm birth among infants with first-trimester INSTI exposure compared to no INSTI exposure [18]. Few studies have compared maternal outcomes such as preeclampsia, gestational diabetes, and laboratory abnormalities between pregnancies exposed to INSTIs versus other ART [19, 20].

The objective of this study was to compare rates of congenital anomalies and adverse maternal and pregnancy outcomes between INSTI-exposed mother-infant dyads and those exposed to PI- or non-nucleoside reverse transcriptase inhibitor (NNRTI) regimens.

## Methods

### Study design

For this retrospective cohort study, electronic health records were abstracted (EHR, Epic systems, Verona, WI) for all pregnant women living with HIV and their infants who were followed by the Children's Hospital Immunodeficiency Program (CHIP) clinic between January 1, 2008 and August 31, 2018. The CHIP clinic is housed within Children's Hospital Colorado, a large tertiary care referral center in Aurora, Colorado. CHIP manages the majority of pregnant women living with HIV in a 7-state region and follows HIV-exposed infants for the first 12–18 months of life. This study was approved by the Colorado Multiple Institutional Review Board (COMIRB) with a waiver of informed consent.

### Data management

Demographic and clinical data were recorded in standardized forms developed in REDCap, hosted by the University of Colorado, Denver [21]. EHR data included all available primary care, obstetric and HIV specialty encounters, inpatient notes and diagnostic test results during the pregnancy period. Race and ethnicity were categorized according to maternal self-report. Spontaneous abortion was defined as fetal death <20 weeks, and stillbirth as death ≥20 weeks gestational age. Small-for-gestational-age (SGA) was defined as weight <10th percentile [22]. Infants with stillbirth or spontaneous abortion were excluded from analyses of congenital anomalies. Twins were excluded from all analyses.

Pregnancies were categorized as having DTG or any INSTI exposure if a mother received these drugs at any time between conception and delivery. ART was categorized by timing of receipt, in the first (gestational age ≤13 weeks), second (14–27), or third (28-delivery) trimesters.

Maternal hematologic and complete metabolic panels were drawn at first visit during pregnancy, within 1 month of initiating or changing ART, and every 1–2 months throughout the duration of pregnancy. Laboratory abnormalities were graded according to the 2017 Division of AIDS Table for Grading the Severity of Adult and Pediatric Adverse Events with pregnancy-specific modifications [23, 24]. For white blood cells, platelets, and absolute neutrophil count, grades were assigned according to the absolute values in the DAIDS table. Grade 2 anemia was defined using a lower limit of hemoglobin according to trimester (9.5 and 8.0 g/dL in trimesters 1 and 2/3, respectively). Grade 3 anemia was defined using a lower limit of

hemoglobin of 7.0 g/dL in all trimesters. For alanine aminotransferase, aspartate aminotransferase, creatinine, bilirubin, and lipase, the upper limits of normal were assigned based on previously published normal ranges in each trimester [24]. Laboratory abnormalities were considered potentially related to ARV exposure if they were identified for the first time at least 3 days after starting a new medication, at any time thereafter while taking the medication, or within 3 days after stopping the medication. "Baseline" viral load and CD4 count were defined by the first value measured during pregnancy, and "delivery" values were obtained within 45 days before or after the date of infant birth (or pregnancy loss).

Congenital anomalies were recorded if they were present at birth and identified in the first year of life. To avoid missing any potential safety signals, we used a broad definition of congenital anomalies. Anomalies were considered major if they required surgical or medical intervention or had serious cosmetic significance, as per the NIAID definition for clinical trials [25]. Genetic screening to identify an etiology of congenital anomalies was performed at the discretion of the treating physicians; infants with a confirmed genetic condition were excluded from analyses of congenital anomalies. Infants whose mothers were seropositive or had unknown serostatus for cytomegalovirus (CMV) or *Toxoplasma gondii* were routinely screened for congenital CMV or *T. gondii* infection [26].

## Statistical analysis

All models fit were binomial family generalized estimating equations (GEEs) with logit link and an exchangeable working correlation structure, which account for correlation in repeat measures among women who had more than 1 pregnancy. This model was used unless otherwise specified. Inferences were conducted using robust standard errors. We compared descriptive characteristics between INSTI- and non-INSTI-exposed mothers and infants using binomial family GEEs for all characteristics except viral load and birth weight, which were evaluated using GEE models with a normal distribution. The lower limit of detection of HIV RNA PCR varied from <50 copies/mL to <20 copies/mL depending upon the year and the institution performing the laboratory test. When calculating median viral load, we used half of the lower limit of detection of HIV RNA PCR for samples with results below the lower limit of detection (i.e., 10 copies/mL for a sample reported as <20 copies/mL). Non-live births were excluded from the descriptive analyses of birth/infant characteristics.

We evaluated the association between DTG or any INSTI exposure in the 1st trimester or the 2nd/3rd trimester and the presence of at least one congenital anomaly (yes/no). The comparison group was either 1st trimester non-INSTI ART exposure or no INSTI exposure, described separately for each result. We evaluated the relationship between INSTI exposure (yes/no) and binary birth outcome (term versus preterm delivery). We also evaluated the relationship between INSTI exposure (yes/no) and pregnancy outcomes including gestational diabetes, SGA, and preeclampsia. We evaluated the association between two demographic variables, race (Black vs. all other races) and maternal age (<18 or >40 vs. 18–40 years old), with preeclampsia, using univariable analyses. Then we performed a multivariable analysis using binomial family GEE that evaluated the association between INSTI exposure (yes/no), maternal age, and preeclampsia.

We used descriptive statistics to report the rates of grade $\geq 2$ and $\geq 3$ maternal chemistry and hematology laboratory abnormalities among women who were and were not exposed to INSTI. For women who were exposed to INSTI, we described laboratory abnormalities both while taking the INSTI, and during other times in the pregnancy. Because the definition of the laboratory adverse events outcome included receipt of an INSTI (or not) at the time of the laboratory abnormality, a statistical comparison of the association between this outcome and INSTI exposure could not be performed.

All analyses were two-tailed, and significance was set at p<0.05. Analyses were performed in SAS 9.4 (SAS Institute).

## Results

### Participant characteristics and antiretroviral regimens

During the study period, 193 mothers living with HIV had 257 pregnancies (**Fig 1**). After excluding 10 twin gestations, 247 pregnancies remained in the analysis. Seventy-seven women received at least 1 INSTI (54 DTG, 14 EVG, 15 RAL), 167 received a PI- or NNRTI- (non-INSTI) regimen, and 3 had missing ART data and were excluded from subsequent analyses. ART was started before conception for 25 women who received an INSTI (13 of whom received DTG) and 87 women who received a non-INSTI regimen.

**Table 1** shows maternal and infant demographics for the 244 pregnancies included in outcome analyses. Women generally had well-controlled HIV, with a median (IQR) CD4 count of 520 (342–686) cells/mm$^3$ and viral load of 69.5 (<20–6967) RNA copies/mL at baseline. Pregnancies exposed to INSTI- vs. non-INSTI regimens did not differ with respect to maternal age, CD4 count and viral load at baseline or delivery, mechanism of delivery (vaginal vs. cesarean delivery), gestational age at delivery, infant race/ethnicity, infant sex, or infant birthweight. No infants acquired HIV.

### Incidence of congenital anomalies

After excluding 2 stillbirths, 2 spontaneous abortions, and one infant with multiple congenital anomalies attributed to Trisomy 21, 76 INSTI-exposed infants (53 of whom were DTG-exposed) and 163 non-INSTI-exposed infants were evaluated for congenital anomalies (**Fig 1**). Overall, 50 congenital anomalies were identified among 36 infants [12 (24%) musculoskeletal; 12 (24%) malformations of the eye, ear, face, or neck; 8 (16%) circulatory; 6 (12%) nervous system; 4 (8%) genital; 3 (6%) urinary; 2 (4%) digestive; 3 (6%) other (all dermatologic); **Table 2**]. No neural tube defects were identified. Twenty-seven (54%) anomalies were major, and 35 (70%) anomalies were diagnoses tracked by the Metropolitan Atlanta Congenital Defects Program (MACDP). None of the infants with congenital anomalies were diagnosed with congenital CMV or *T. gondii* infection.

### Association of congenital anomalies with antiretroviral drugs

Among 40 infants exposed to an INSTI in the first trimester, 11 (27.5%) had at least one congenital anomaly compared with 13 (11.9%) among 109 infants exposed to non-INSTI ART in the first trimester (OR = 2.61; 95%CI = 1.15–5.94; **Fig 2**). When anomalies were restricted to those recognized by the MACDP, the rates of congenital anomalies were 17.5% (7 out of 40) among infants with first trimester INSTI exposure vs. 8.6% (14 out of 163) among infants with no INSTI exposure (OR = 2.23; 95%CI = 0.89–5.56). Infants whose INSTI exposure began in the second or third trimesters did not have increased odds of congenital anomalies; 6 out of 36 (16.7%) of these infants had an anomaly compared with 18 out of 163 (11.0%) among infants with no INSTI exposure (OR = 1.58; 95%CI = 0.58–4.30). The anomaly types that occurred most frequently among INSTI-exposed infants were disorders of the eye, ear, face, neck, and musculoskeletal systems (**Fig 3**).

Among 24 infants exposed to DTG in the first trimester, 7 (29.2%) had congenital anomalies compared with 13 (11.9%) among 109 infants exposed to non-INSTI ART in the first trimester (OR = 2.55; 95%CI = 1.07–6.10; **Fig 2**). When anomalies were restricted to those recognized by the MACDP, the rates of congenital anomalies were 16.7% (4 out of 24) among

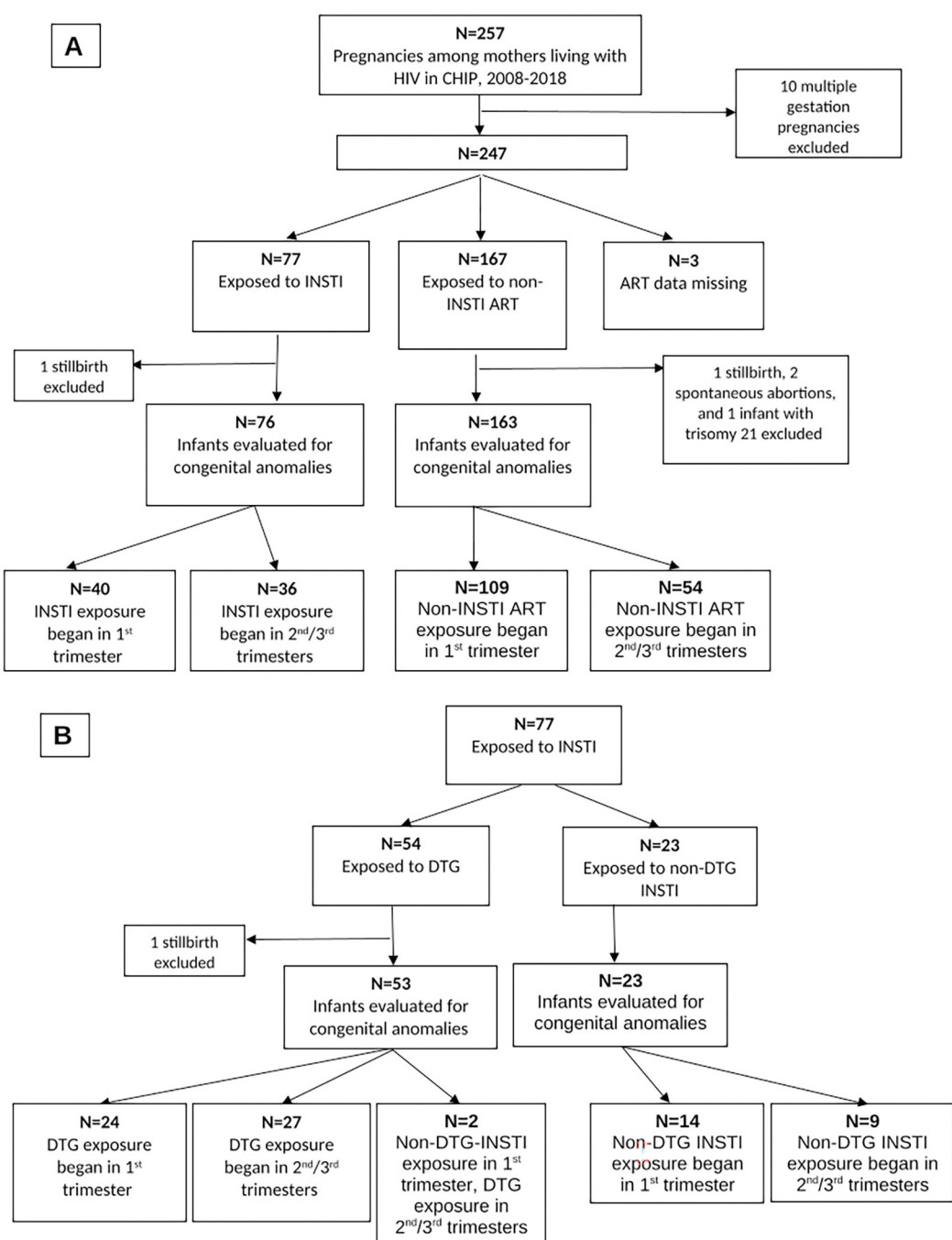

**Fig 1. Flowchart of HIV-exposed infants followed in the Children's Hospital Immunodeficiency Program (CHIP) clinic who were included in this study.** A) All infants, divided by type of maternal antiretroviral (ART) exposure; B) infants exposed to maternal integrase inhibitors (INSTI), divided by exposure or not to dolutegravir (DTG).

infants with first trimester DTG exposure vs. 8.6% (14 out of 163) among infants with no INSTI exposure (OR = 2.14; 95%CI = 0.73–6.25). Infants whose DTG exposure began in the second or third trimesters did not have increased odds of congenital anomalies; 5 out of 29 (17.2%) of these infants had an anomaly compared with 18 out of 163 (11.0%) among infants with no INSTI exposure (OR = 1.80; 95%CI = 0.61–5.30).

**Table 1. Demographic and HIV disease characteristics of INSTI- and non-INSTI-exposed mothers and infants.**

| Maternal Characteristic | | INSTI-exposed (n = 77) | Non-INSTI-exposed (n = 167) | P |
|---|---|---|---|---|
| Maternal age at delivery/outcome (years) | Median (IQR) | 31 (27–34) | 30 (26–35) | 0.15 |
| Maternal CD4 count at baseline (cells/mm³) [a] | Median (IQR) | 531 (302–778) | 517 (342–651) | 0.66 |
| Maternal CD4 count at delivery (cells/mm³) [a] | Median (IQR) | 520 (328–706) | 570 (408–731) | 0.36 |
| Maternal viral load at baseline (copies/mL) [a] | Median (IQR) | 121 (<20–15960) | 71.5 (<20–6445) | 0.74 |
| Maternal viral load at delivery (copies/mL) [a] | Median (IQR) | <20 (<20-<20) | 24 (<20–39) | 0.47 |
| Trimester at start of maternal ART | Preconception | 25 (32.5%) | 87 (52.1%) | 0.20 |
| | 1st trimester | 15 (19.5%) | 26 (15.6%) | |
| | 2nd or 3rd trimester | 37 (48.1%) | 54 (32.3%) | |
| Toxicology screen ever positive during pregnancy | Yes | 10 (12.5%) | 16 (9.2%) | 0.26 |
| Birth/Infant Characteristic [b] | | INSTI-exposed (n = 76) | Non-INSTI-exposed (n = 164) | P |
| Mechanism of delivery [a] | Vaginal | 37 (48.68%) | 84 (51.22%) | 0.46 |
| | Cesarean | 38 (50.00%) | 77 (46.95%) | |
| Gestational age at delivery (weeks) | Median (IQR) | 38 (37–39) | 39 (37–39) | 0.44 |
| Infant birth weight (grams) [a] | Median (IQR) | 2998 (2661–3415) | 3000 (2700–3221) | 0.85 |
| Infant sex | Male | 37 (48.68%) | 83 (50.61%) | 0.83 |
| Infant race [a] | Black | 46 (60.53%) | 86 (52.44%) | 0.58 |
| | White | 21 (27.63%) | 52 (31.71%) | |
| | Other | 4 (5.26%) | 11 (6.71%) | |
| Infant ethnicity [a] | Hispanic | 13 (18.06%) | 48 (30.77%) | 0.06 |

[a] Variables with missing data for INSTI-exposed vs. non-INSTI-exposed infants, respectively: maternal CD4 count at baseline (3, 9); maternal CD4 count at delivery (25, 56); maternal viral load at baseline (1, 1); maternal viral load at delivery (8, 19); mode of delivery (1, 3); infant birth weight (0, 2); infant race (5, 15); infant ethnicity (4, 8). All proportions are calculated among those with non-missing data.

[b] Non-live births (n = 4) were excluded from all birth/infant characteristics.

INSTI, integrase inhibitor; IQR, interquartile range

One stillbirth that had been exposed to DTG beginning in the 2nd trimester had dysmorphic features noted at autopsy, including microcephaly, low set ears, retrognathia, and elongated fingers. Cytogenetic studies of fetal tissue revealed no abnormalities of chromosome number or structure. This mother was diagnosed with HIV in the 2nd trimester of pregnancy and received 5 weeks of DTG prior to presenting in preterm labor. The 3 remaining non-live births (1 stillbirth and 2 spontaneous abortions) occurred after non-INSTI ART exposures. All 3 fetuses were anatomically normal, as documented by fetal autopsies in two cases and gross examination in the remaining case.

## Association of maternal and pregnancy outcomes with antiretroviral drugs

We examined maternal and pregnancy outcomes with respect to exposure to INSTI- vs. non-INSTI ART (Table 3). We found an association between INSTI exposure at any time during pregnancy and preeclampsia (OR = 4.73; 95%CI = 1.70–13.19). To evaluate for potential confounders, we examined the relationship of maternal age and race with preeclampsia in univariable analyses. We only found an association between maternal age and preeclampsia (OR = 1.11; 95% CI = 1.02–1.19). In a multivariable model including maternal age, INSTI exposure during pregnancy remained a significant predictor of preeclampsia (p = 0.01). Of note, four women had pre-existing chronic hypertension (two women received INSTI-containing ART and each had one pregnancy, two women received non-INSTI ART and had a total of 4 pregnancies); these numbers were too small to include hypertension as a co-variable in the multivariable model of preeclampsia. Other maternal and pregnancy outcomes did not

**Table 2. Description of specific congenital anomalies that occurred among live-born infants of single gestation pregnancies, by organ system.**

| Anomaly type | Specific Diagnoses | Multisystem anomalies[a] | Major[b] | MACDP[c] | Earliest maternal ART in pregnancy | Gestational age (weeks) at start of maternal ART |
|---|---|---|---|---|---|---|
| Circulatory System | Atrial septal defect | | Major | MACDP | elvitegravir, cobicistat, emtricitabine, tenofovir | preconception |
| | Atrial septal defect | | | MACDP | lopinavir, ritonavir, lamivudine, zidovudine | preconception |
| | Innominate artery compression syndrome | | Major | MACDP | atazanavir, ritonavir, emtricitabine, tenofovir | 13 (1st trimester) |
| | Left ventricle noncompaction | Multi | Major | MACDP | atazanavir, ritonavir, lamivudine, abacavir | 15 |
| | Patent foramen ovale and patent ductus arteriosus | | | MACDP | rilpivirine, emtricitabine, tenofovir | preconception |
| | Pulmonic valve dysplasia and stenosis | | | MACDP | none | N/A |
| | Ventricular septal defect | Multi | | MACDP | dolutegravir, lamivudine, abacavir | preconception |
| | Ventricular septal defect | | | MACDP | elvitegravir, cobicistat, emtricitabine, tenofovir | preconception |
| Digestive System | Hirschsprung's disease | Multi | Major | MACDP | atazanavir, ritonavir, emtricitabine, tenofovir | preconception |
| | Ilial volvulus due to abnormal mesentery development with congenital band | | Major | MACDP | dolutegravir, lamivudine, abacavir | preconception |
| Eye, Ear, Face, and Neck | Ankyloglossia | | Major | MACDP | atazanavir, lamivudine, zidovudine | preconception |
| | Ankyloglossia | Multi | | MACDP | efavirenz, emtricitabine, tenofovir | preconception |
| | Ankyloglossia and laryngomalacia | | Major | MACDP | dolutegravir, emtricitabine, tenofovir | 23 |
| | Bilateral nasolacrimal duct obstruction | | Major | | elvitegravir, cobicistat, emtricitabine, tenofovir | preconception |
| | Congenital dacryocystocele | | Major | MACDP | atazanavir, ritonavir, emtricitabine, tenofovir | preconception |
| | Laryngomalacia | | | | atazanavir, ritonavir, lamivudine, abacavir | 7 (1st trimester) |
| | Laryngomalacia and tracheomalacia | | Major | | atazanavir, ritonavir, emtricitabine, tenofovir | preconception |
| | Middle ear dysfunction resulting in conductive hearing loss | | Major | MACDP | dolutegravir, lamivudine, abacavir | preconception |
| | Strabismus | | Major | | dolutegravir, emtricitabine, tenofovir | 25 |
| | Strabismus | Multi | Major | | dolutegravir, emtricitabine, tenofovir | 8 (1st trimester) |

(*Continued*)

**Table 2.** (Continued)

| Anomaly type | Specific Diagnoses | Multisystem anomalies[a] | Major[b] | MACDP[c] | Earliest maternal ART in pregnancy | Gestational age (weeks) at start of maternal ART |
|---|---|---|---|---|---|---|
| | Tracheomalacia | | | | rilpivirine, emtricitabine, tenofovir | 10 (1st trimester) |
| | Tracheomalacia | Multi | | | elvitegravir, cobicistat, emtricitabine, tenofovir | 18 |
| Genital Organs | Penile torsion | | Major | MACDP | atazanavir, ritonavir, emtricitabine, tenofovir | preconception |
| | Undescended testes, bilateral, self-resolved | Multi | | MACDP | dolutegravir, lamivudine, abacavir | preconception |
| | Undescended testicle, unilateral, self-resolved | | | MACDP | lopinavir, ritonavir, lamivudine, zidovudine | 24 |
| | Undescended testes | Multi | Major | MACDP | atazanavir, ritonavir, emtricitabine, tenofovir | preconception |
| Musculoskeletal System | Acetabular dysplasia with leg length discrepancy | | Major | MACDP | efavirenz, emtricitabine, tenofovir | 1 (1st trimester) |
| | Arthrogryposis and talipes equinovarus | Multi | Major | MACDP | dolutegravir, lamivudine, abacavir | preconception |
| | Butterfly vertebrae | Multi | | MACDP | raltegravir, emtricitabine, tenofovir | 12 (1st trimester) |
| | Developmental hip dysplasia | | | | dolutegravir, lamivudine, abacavir | 24 |
| | Developmental hip dysplasia | Multi | Major | | efavirenz, emtricitabine, tenofovir | preconception |
| | Diplegia with limb dystonia | | Major | | lopinavir, ritonavir, lamivudine, abacavir | preconception |
| | Pseudoarthrosis of clavicle | Multi | Major | MACDP | elvitegravir, cobicistat, emtricitabine, tenofovir | preconception |
| | Syndactyly | Multi | | MACDP | atazanavir, ritonavir, lamivudine, abacavir | 15 |
| | Talipes equinovarus | Multi | Major | MACDP | atazanavir, ritonavir, emtricitabine, tenofovir | preconception |
| | Umbilical hernia | Multi | | | dolutegravir, emtricitabine, tenofovir | 8 (1st trimester) |
| | Umbilical hernia | | | | atazanavir, ritonavir, emtricitabine, tenofovir | 18 |
| | Umbilical hernia | Multi | | | dolutegravir, lamivudine, abacavir | 12 (1st trimester) |
| Nervous System | Agenesis of the corpus callosum | Multi | Major | MACDP | atazanavir, ritonavir, emtricitabine, tenofovir | preconception |
| | Macrocephaly and developmental delay | | Major | MACDP | atazanavir, ritonavir, emtricitabine, tenofovir | 11 (1st trimester) |

(*Continued*)

Table 2. (Continued)

| Anomaly type | Specific Diagnoses | Multisystem anomalies[a] | Major[b] | MACDP[c] | Earliest maternal ART in pregnancy | Gestational age (weeks) at start of maternal ART |
|---|---|---|---|---|---|---|
| | Macrocephaly | Multi | | MACDP | elvitegravir, cobicistat, emtricitabine, tenofovir | 18 |
| | Macrocephaly | | | MACDP | lopinavir, ritonavir, lamivudine, zidovudine | 15 |
| | Microcephaly and developmental delay | Multi | Major | MACDP | atazanavir, ritonavir, lamivudine, abacavir | 15 |
| | Rhomboencephalosynapsis, thin corpus callosum, prominent ventricles and extra-axial spaces, suggestive of diffuse atrophy | Multi | Major | MACDP | raltegravir, emtricitabine, tenofovir | 12 (1st trimester) |
| Urinary System | Duplicate right renal collecting system | Multi | | MACDP | elvitegravir, cobicistat, emtricitabine, tenofovir | preconception |
| | Hydronephrosis | | | MACDP | atazanavir, ritonavir, emtricitabine, tenofovir | 9 (1st trimester) |
| | Vesicoureteral reflux | Multi | Major | MACDP | atazanavir, ritonavir, emtricitabine, tenofovir | preconception |
| Other | Hemangioma | Multi | Major | | dolutegravir, lamivudine, abacavir | 12 (1st trimester) |
| | Hemangioma | | | | atazanavir, ritonavir, lamivudine, abacavir | preconception |
| | Sebaceous nevus with associated alopecia on scalp | | | MACDP | efavirenz, lamivudine, tenofovir | 18 |

[a] Anomalies that occurred in infants with more than one anomaly type are indicated as having multisystem anomalies. Fifty anomalies occurred among 36 infants.

[b] Major anomalies were defined as those that required surgical or medical intervention or had serious cosmetic significance.

[c] MACDP, Metropolitan Atlanta Congenital Defects Program. "Yes" indicates that the diagnosis is tracked by the MACDP, and "no" indicates that the diagnosis is considered a minor defect, a normal variant, or another type of condition that is not tracked by MACDP.

Abbreviations: ART, antiretroviral treatment.

differ between groups, including preterm birth, which occurred in 16.88% of INSTI-exposed and 13.77% of non-INSTI exposed pregnancies.

Of 77 women who received INSTIs, 11 (14.3%) had a grade ≥2 chemistry or hematology laboratory abnormality while taking the INSTI, and 13 (16.9%) had a grade ≥2 laboratory abnormality at another time during the pregnancy. Of 167 women who received non-INSTI ART, 47 (28.1%) had a grade ≥2 laboratory abnormality during pregnancy. The most common grade ≥2 laboratory abnormalities among women receiving INSTIs included hyperbilirubinemia (n = 6), elevated lipase (n = 2), elevated aspartate aminotransferase (n = 2), and anemia (n = 2). The most common grade ≥2 laboratory abnormalities among women receiving non-INSTI ART included hyperbilirubinemia (n = 65), elevated lipase (n = 21), and thrombocytopenia (n = 10).

Of 77 women who received INSTIs, 2 (2.6%) had a grade ≥3 chemistry or hematology laboratory abnormality while taking the INSTI, and 3 (3.9%) at another time during the pregnancy. Of 167 women who received non-INSTI ART, 27 (16.2%) had a grade ≥3 laboratory abnormality. The grade ≥3 laboratory abnormalities among women receiving INSTIs included anemia (n = 2), elevated lipase (n = 1), and elevated serum creatinine (n = 1). The most common

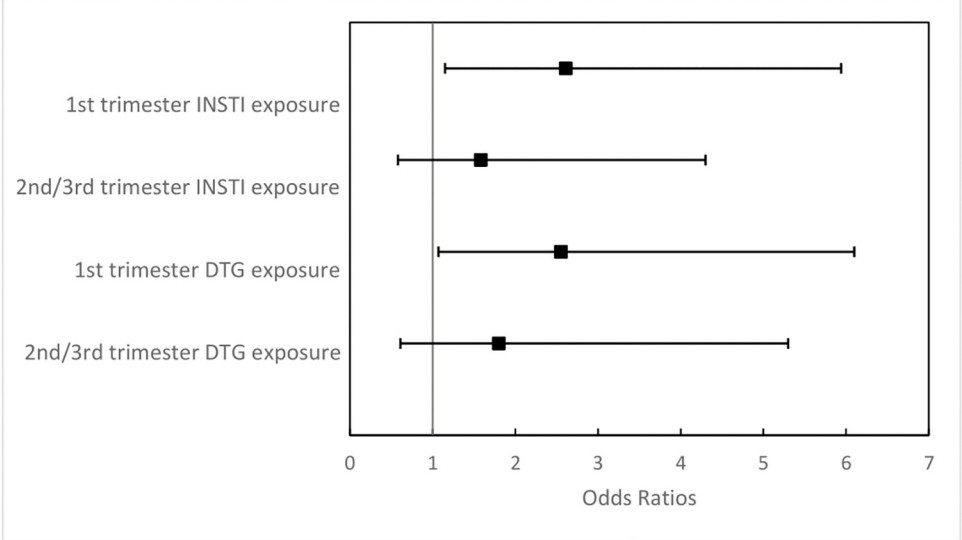

**Fig 2. Association between in utero exposure to any integrase inhibitor (INSTI) or dolutegravir (DTG) and the odds of having a congenital anomaly.** The reference group for first trimester DTG or any INSTI exposure is infants with first trimester non-INSTI ART exposure. The reference group for 2nd/3rd trimester DTG or any INSTI exposure is infants never exposed to INSTI ART.

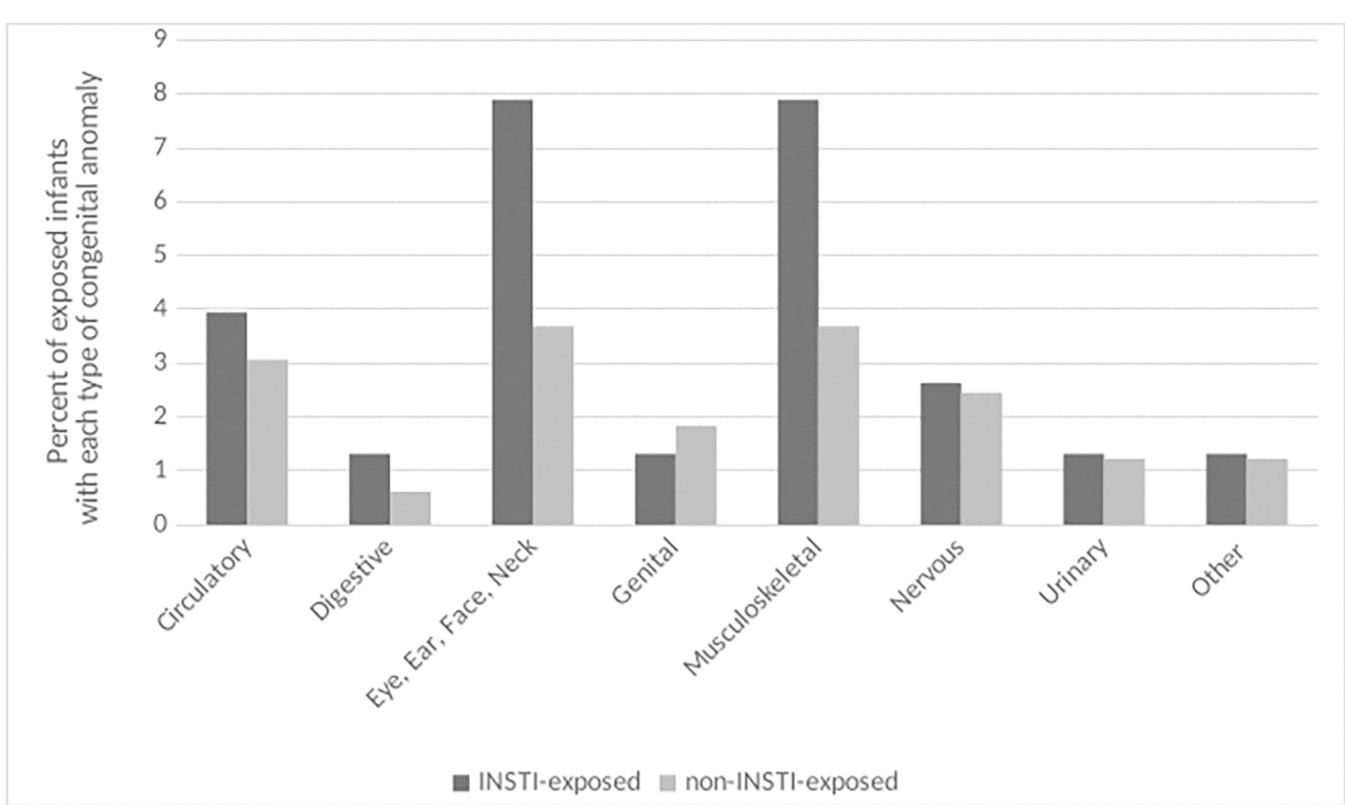

**Fig 3. Percentage of infants ever exposed to an integrase inhibitor (INSTI) or non-INSTI regimen with each type of congenital anomaly.**

**Table 3. Pregnancy outcomes of women receiving INSTI- and non-INSTI antiretroviral therapy.**

| Outcome | | INSTI-exposed (n = 77) | Non-INSTI-exposed (n = 167) | P [a] |
|---|---|---|---|---|
| Birth outcome | Term delivery | 63 (81.82%) | 141 (84.43%) | 0.37 |
| | Preterm delivery | 13 (16.88%) | 23 (13.77%) | |
| | Spontaneous abortion | 0 (0%) | 2 (1.20%) | |
| | Stillbirth | 1 (1.30%) | 1 (0.60%) | |
| Small for gestational age [b,c] | | 14 (18.92%) | 27 (18.00%) | 0.81 |
| Preeclampsia | | 11 (14.29%) | 6 (3.59%) | <0.01 |
| Gestational diabetes | | 6 (7.79%) | 16 (9.58%) | 0.39 |

[a] Birth outcome, small for gestational age, preeclampsia, and gestational diabetes were compared using binomial family generalized estimating questions (GEEs).

[b] Variables with missing data for women receiving INSTI- vs. non-INSTI ART, respectively: small for gestational age (2, 14). All proportions are calculated among those with non-missing data.

[c] Non-live births (n = 4) were excluded from the following characteristics: small for gestational age

INSTI, integrase inhibitor

grade $\geq 3$ laboratory abnormalities among women receiving non-INSTIs included hyperbilirubinemia (n = 37), anemia (n = 4), and elevated lipase (n = 2).

## Discussion

In this single-center cohort over a 10-year period, we identified a more than two-fold higher risk of congenital anomalies among infants exposed to INSTI- vs. non-INSTI ART during the first trimester of pregnancy. Of the 40 INSTI exposures in the first trimester, 24 included DTG, which on its own was associated with a more than two-fold increased risk of anomalies. However, when evaluating all INSTI exposures together, the association with anomalies remained significant, suggesting that a class effect may exist. The birth defects we identified among INSTI-exposed infants did not cluster by organ system, although most anomalies observed among INSTI-exposed infants were either musculoskeletal or otolaryngologic. We did not identify any neural tube defects.

To our knowledge, this is the first study showing a significant association between a variety of congenital anomalies and the use of INSTI in the first trimester of pregnancy. Several previous studies have focused on the specific association between DTG and neural tube defects [2, 3, 5]. Recent publications describing other types of congenital anomalies among INSTI-exposed infants are subject to limitations such as small sample sizes, lack of a control group, and/or lack of a true denominator [4, 8, 9, 11–14]. However, at least two large observational studies identified non-significant trends toward increased congenital anomalies among infants exposed to INSTI in the first trimester [10, 15]. Ideally, the potential associations we identified should be confirmed in additional studies with larger sample sizes.

We did not find significant associations between congenital anomalies and the use of INSTI in the second or third trimester. This aligns with what is known about the timing of organogenesis, which is complete by 12–14 weeks of gestation. In support of the hypothesis that in utero INSTI exposure could trigger abnormal fetal development, one research group has presented preliminary data showing that cultured human embryonic stem cells experience apoptosis, reduction in viability and/or loss of pluripotency after in vitro exposure to various drugs from the INSTI class at therapeutic and sub-therapeutic concentrations [27, 28].

We noted high overall rates of congenital anomalies in our cohort: 27.5% in INSTI-exposed and 11.9% in non-INSTI-exposed infants. In comparison, the MACDP estimates that major birth defects occur in approximately 3% of US births [29]. Previous studies of HIV- and/or

ART-exposed infants have documented rates of congenital anomalies ranging from 5.5 to 14.8%, suggesting that birth defects may occur more frequently in this population [30–32]. One pharmacokinetics study of DTG in pregnant women diagnosed anomalies in 24% of exposed infants, similar to the proportion we identified [9]. Importantly, surveillance programs that depend on registry data are subject to bias from under-reporting/differential reporting, under-ascertainment/differential ascertainment of birth defects, and loss to follow up. For example, the MACDP only captures defects identified upon surface examination at the time of delivery. We suspect that our clinic's practice of following HIV-exposed infants with frequent visits and careful examinations over the first 12–18 months of life allowed us to identify anomalies that may have been overlooked in other settings, including anomalies that are not apparent upon surface examination. We do not believe that our elevated rate of congenital anomalies is attributable to the inclusion of inappropriate diagnoses. Seventy percent of the anomalies we identified are diagnoses that are tracked by the MACDP (see **Table 2**), and diagnoses that are not tracked by the MACDP have been included in other reports of the association between congenital anomalies and INSTI exposure [8, 15, 33, 34]. Of note, congenital anomalies have been associated with maternal pre-gestational diabetes; only two women in this cohort had a diagnosis of pre-gestational diabetes and neither of their infants had an anomaly.

We identified higher rates of preeclampsia in women who received INSTI vs. non-INSTI ART during pregnancy. A previous study reported higher rates of gestational hypertension among women receiving DTG compared to efavirenz, although rates of hypertension were lower than among women without HIV in that study [19]. Other studies have identified a relationship between INSTIs and adverse cardiometabolic outcomes, with one study in non-pregnant women showing increases in both blood pressure and HbA1c in women who switched from a PI- or NNRTI to an INSTI regimen, compared with women who did not switch [35]. However, another study found a protective effect of DTG against gestational diabetes [36]. We did not find an association between INSTI exposure and any other adverse maternal and/or pregnancy outcomes, including gestational diabetes, SGA, spontaneous abortion, stillbirth, or preterm birth. Other studies have identified an association between INSTI exposure during pregnancy and preterm birth [18]. Of note, our comparison group included women who received PIs, which are associated with increased rates of preterm birth [37, 38]; this may have limited our ability to detect an association between INSTI exposure and preterm birth, should one exist.

We noted that women on INSTI had fewer grade ≥3 chemistry and hematology laboratory abnormalities compared to women on non-INSTI regimens, although this could not be compared in a formal statistical analysis. Hyperbilirubinemia made up the majority of laboratory abnormalities in our cohort. Of note, we used pregnancy-specific normal ranges for bilirubin in the current study, which are lower than normal ranges in nonpregnant adults [24, 39].

We did not identify differences in maternal viral load at delivery between mothers receiving INSTI vs. non-INSTI regimens. Notably, most women in our cohort had well-controlled HIV at baseline. One benefit of INSTIs is rapid reduction in viral load, which makes their use beneficial in particular scenarios, i.e. pregnant women who present late to care. Several studies have identified lower viral loads at delivery among pregnant women receiving INSTI compared with non-INSTI regimens [17, 34, 40]. Other potential benefits of INSTI include a low side effect profile and high genetic barrier to resistance. Our findings suggest that these potential benefits must be weighed against the potential risks of congenital anomalies and preeclampsia associated with INSTI use in pregnancy.

Limitations of this study include its retrospective nature. The total number of pregnancies included in the study was relatively small, which resulted in wide confidence intervals around

our odds ratios. Our small sample size also precluded the ability to perform multivariable analyses; however, most characteristics were well balanced between infants exposed to INSTI and non-INSTI regimens. In addition, this study was performed at a single site, which may limit the generalizability of our results. The introduction of INSTIs overlapped with changes in the nucleoside reverse transcriptase inhibitor (NRTI) backbone from zidovudine- to abacavir- or tenofovir-containing regimens, such that we were unable to match or control for the use of NRTI. Although one recent large study did not find any association between specific NRTIs and congenital anomalies, previous studies attributed fetal cardiac malformations to in utero zidovudine exposure [31, 41–43]. However, only one out of eight infants with a cardiac anomaly in our study was exposed to zidovudine. The temporal change in NRTI backbone may also explain the higher incidence of grade ≥3 toxicities in the non-INSTI cohort and it may have resulted in additional confounders. We did not collect data on maternal folate intake, but women in the CHIP program are encouraged to take prenatal vitamins, and grains in the U.S. are supplemented with folate. Our ability to perform multivariable analyses of the preeclampsia outcome was limited by the fact that we were unable to collect data on maternal body mass index, tobacco or alcohol use, or medications other than ART. However, few women in our cohort had other risk factors associated with preeclampsia, such as antiphospholipid antibody syndrome (n = 0), pregestational diabetes (n = 2), or assisted reproductive technology (n = 0) [44].

In conclusion, we identified a potential association between first trimester INSTI exposure and congenital anomalies of the non-neural tube defect type. We also identified an association between receipt of INSTI during pregnancy and preeclampsia. Ongoing surveillance of women and infants with gestational INSTI exposure is necessary to confirm these potential safety concerns.

## Acknowledgments

CHIP Perinatal Team members who contributed to this study and/or publication: Megan Dinnebeil[1], Jennifer Dunn[2], Phillip Ferrero[1], Kay Kinzie[1], Elizabeth J. McFarland[2]*, Jennifer Moor[1], Kacey Navarro[2], and Suzanne Paul[2].

[1]Children's Hospital Colorado, Aurora, CO, USA

[2]Department of Pediatrics, University of Colorado, Aurora, CO, USA

*Medical Director of CHIP; email: Betsy.McFarland@cuanschutz.edu

## Author Contributions

**Conceptualization:** Adriana Weinberg.

**Data curation:** Christiana Smith, Joyce F. Sung, Jennifer R. McKinney, Kirk B. Fetters, Sarah Lazarus, Shannon Capraro, Emily Barr, Carrie Glenny, Jenna Buehler, Adriana Weinberg.

**Formal analysis:** Angela J. Fought.

**Writing – original draft:** Christiana Smith.

**Writing – review & editing:** Joyce F. Sung, Jennifer R. McKinney, Torri D. Metz, Kirk B. Fetters, Sarah Lazarus, Shannon Capraro, Emily Barr, Carrie Glenny, Jenna Buehler, Adriana Weinberg.

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
