## [Decision Letter · Decision Letter 0]

22 Nov 2022

PONE-D-22-27763Congenital Malformations and Preeclampsia Associated with Integrase Inhibitor Use in PregnancyPLOS ONE

Dear Dr. Smith,

Thank you for submitting your manuscript to PLOS ONE. After careful consideration, we feel that it has merit but does not fully meet PLOS ONE’s publication criteria as it currently stands. Therefore, we invite you to submit a revised version of the manuscript that addresses the points raised during the review process.

Please comply with Reviewers’ comments. In particular, recognize the limitations of your work and check the title.

We look forward to receiving your revised manuscript.

Kind regards,

Carlo Torti

Academic Editor

PLOS ONE

Journal Requirements:

   "I have read the journal's policy and the authors of this manuscript have the following competing interests: Adriana Weinberg discloses research grants from GlaxoSmithKline and Merck that were paid to the University of Colorado and personal fees from Merck, Squirus and GlaxoSmithKline for work unrelated to this study. The remaining authors have declared that no competing interests exist. "

5. One of the noted authors is a group or consortium [CHIP Perinatal Team]. In addition to naming the author group, please list the individual authors and affiliations within this group in the acknowledgments section of your manuscript. Please also indicate clearly a lead author for this group along with a contact email address.

Additional Editor Comments:

I agree with the Reviewers that the paper has merits but limitations should be recognized.

Reviewers' comments:

Reviewer's Responses to Questions

**Comments to the Author**

1. Is the manuscript technically sound, and do the data support the conclusions?

Reviewer #1: Partly

Reviewer #2: Yes

Reviewer #3: Yes

2. Has the statistical analysis been performed appropriately and rigorously? 

Reviewer #1: N/A

Reviewer #2: Yes

Reviewer #3: I Don't Know

3. Have the authors made all data underlying the findings in their manuscript fully available?

Reviewer #1: No

Reviewer #2: Yes

Reviewer #3: Yes

4. Is the manuscript presented in an intelligible fashion and written in standard English?

Reviewer #1: Yes

Reviewer #2: Yes

Reviewer #3: Yes

5. Review Comments to the Author

Reviewer #1: This is a retrospective single-center (US) cohort study over a 10-year period, which reports higher rate of congenital anomalies associated with exposure to integrase inhibitors in the first trimester compared to exposure to non-INSTI ART in the first trimester.

The main limitation of this article is that overall percentage of congenital anomalies is very high, even in non-INSTI-exposed infants. It would be important to provide baseline data for the US population and to make sure the categories used to report congenital anomalies in this paper meet standards for this kind of report.

Conclusion from this publication should be very careful as they are very different from previous reports. For instance, in a previous publication cited by this paper (Money 2019), there were 80 cases with dolutegravir exposure in the first trimester (including 69 cases with dolutegravir at conception) with four cases of non-chromosomal congenital anomalies, giving a rate of 5.0% (95% CI 1.4–12.3%) – this is highly different from this work.

Comparison of the group with exposure to integrase inhibitors in the first trimester to the group of exposure to non-INSTI ART in the first trimester is already done, but comparison to the group of no exposure to ART in the first trimester (n=90) should be added, while comparison to group never exposed to an INSTI should be removed.

Further comments:

-Congenital anomalies should be categorized using the ICD https://icd.who.int/browse10/2016/en#/XVII.

-Still birth should not be excluded from the analyses regarding congenital anomalies as autopsy data are available.

-details the type of ART used in the non-INSTI group (what kind of a PI or NNRTI?).

-How was defined race/ethnicity in children and mothers ?

-Were the 36 infants with congenital anomalies tested for common causes of such anomalies, such as congenital CMV infection and genetic conditions? Of note, I am not sure that the trisomy 21 case should be excluded – either exclude all genetic conditions or none.

- Laboratory abnormalities: what is the protocol in place to follow lab toxicity in pregnant women living with HIV? How frequently were these lab tests performed in pregnancy in each group?

-Preeclampsia results: several other confounding factors should be addressed (BMI, parity, smoking…) – not sure you have enough cases for this.

-Table 1: Add gestational age at ART initiation (preconception versus first trimester versus 2 and 3rd trimester); Viral load at baseline and at delivery should be reported in caterogies (undetectable versus detectable)

-Table 2: Each infant should be represented only once, with a new category ‘Multisystem anomalies’ for those who have several anomalies; ; change also this in the text and in the abstract.

-Table 3: Remove Polyhydramnnios, Oligohydramnios Placenta previa - Placental abruption (not relevant)

Reviewer #2: Thank you for the opportunity to review this manuscript which is well written.

The manuscript describes associations between timing of INSTI-and non-INSTI ART on congenital anomalies in infants as well as other pregnancy outcomes, finding increased rates of congenital anomalies in infants born to women receiving INSTI-based ART pre-conception and in the first trimester, as well as increased pre-eclampsia.

Some minor revisions suggested:

Introduction:

-Line 45 suggest adding maternal health benefit to maternal ART (not just PMTCT)

-Line 61-63 suggest explaining clearly the issue with " lack of denominators" upfront, since this comes up a number of times in the manuscript but may not be understood by all and is unclear as written here. This becomes clearer in the sentence lines 70 onwards, but suggest that this explanation is included upfront.

- Suggest adding some information on background congenital anomaly rate in the community where the study took place

Methods:

- Please include a brief standard of care explanation, with inclusion of whether women who are periconception receive folate supplementation in the periconception/1st TM through pregnancy? Also, whether there is grain fortification with folate? This could be a potential factor related to no NTD being seen?

- Line 116-117 is not clear to me, specifically ".....starting a new medication or within 3 days of stopping a new medication". Should this be resolves within 3 days of stopping? That seems too short though? Please clarify

- Obesity is an important factor related to congenital anomalies, particularly cardiac anomalies. Could the BMI not have been calculated using available weight and height, understanding that pre-conception weight may not have been available, but there is data from the first TM in many women.

Table 2:

- This table is quite long-consider highlighting 1st TM exposures as well, rather than just pre-conception

Discussion

- Line 294-296: point well taken, but perhaps also specify that many of the pregnancy registries collect surface examination data only at delivery/first few weeks and have the advantage of larger numbers, but do not investigate for internal anomalies that are not visible.

- Obesity data not available is mentioned as a limitation, although please see previous comments regarding calculation which should be done if possible.

Reviewer #3: The paper is interesting but as the authors point out in the study limitations the cohort has many limitations, not the least of which is the failure to report alcohol and smoking for example

To make it definitively acceptable the authors could check only major congenital defects, in particular major anomalies were defined as those that required surgical or medical intervention or had serious cosmetic significance.

Also with regard to the diagnosis of preeclampsia, risk factors are missing: antiphospholipid antibody syndrome, Chronic hypertension, Pregestational diabetes and body mass index (BMI) >30 and assisted reproductive technology. (Bartsch E, Medcalf K E, Park A L, Ray J G. Clinical risk factors for pre-eclampsia determined in early pregnancy: systematic review and meta-analysis of large cohort studies BMJ 2016; 353 :i1753 doi:10.1136/bmj.i1753).

Finally, if the authors are able to complete these requirements, it would also be helpful not to overemphasize the role of INSTIs in the title, but simply report the results of their single-center cohort with many limitations.

6. PLOS authors have the option to publish the peer review history of their article (what does this mean?). If published, this will include your full peer review and any attached files.

Reviewer #1: No

Reviewer #2: No

Reviewer #3: No

---

## [Author Response · Author response to Decision Letter 0]

23 Jan 2023

We appreciate the thoughtful reviews and believe that we have responded to all reviewer concerns. Please note that all page lines reference the track changes version of the manuscript file. 

Editor Comments:

The figure files have been revised as requested. 

The original version of our manuscript includes this sentence: “This study was approved by the Colorado Multiple Institutional Review Board (COMIRB) with a waiver of informed consent.” (lines 99-100) We believe this statement satisfies the requirement. 

 "I have read the journal's policy and the authors of this manuscript have the following competing interests: Adriana Weinberg discloses research grants from GlaxoSmithKline and Merck that were paid to the University of Colorado and personal fees from Merck, Squirus and GlaxoSmithKline for work unrelated to this study. The remaining authors have declared that no competing interests exist. "

We confirm that the competing interests we described do not alter our adherence to PLOS ONE policies on sharing data and materials. Please update the online submission form on our behalf. 

The underlying data used to reach the conclusions drawn in the manuscript derive from human research participant data. We do not feel that it is ethically appropriate to publicly share this data set because it contains potentially identifiable information from a small group of vulnerable participants from a single institution. Even after removing identifiers such as name, address, and dates, the individuals included in this data set may still be identifiable because of their physical location in Colorado coupled with their unique medical diagnoses. Because all of the individuals in this data set are living with HIV, we feel that it is especially important to protect their identities and avoid unwanted disclosure of this diagnosis. Rather than share our data set publicly, we are happy to make it available upon request. However, we would like to point out that our institution does not have an established point of contact to field external requests for access to this sensitive data set (ie a data access committee, ethics committee, or other institutional body). Investigators interested in accessing the data would need to work with the University of Colorado Office of Grants and Contracts to develop a MTA to obtain the intellectual property. However, the Office of Grants and Contracts would need to coordinate with a member of the study team (either the first author, senior author, or statistician) for access to the data. We are happy to list these authors’ names as the points of contact, but we see that this is discouraged by PLOS One. We would welcome the editor’s advice regarding how best to proceed. 

5. One of the noted authors is a group or consortium [CHIP Perinatal Team]. In addition to naming the author group, please list the individual authors and affiliations within this group in the acknowledgments section of your manuscript. Please also indicate clearly a lead author for this group along with a contact email address.

The original version of our manuscript contained the following information in the Acknowledgments section: “CHIP Perinatal Team members who contributed to this study and/or publication: Megan Dinnebeil, Jennifer Dunn, Phillip Ferrero, Kay Kinzie, Elizabeth J. McFarland, Jennifer Moor, Kacey Navarro, and Suzanne Paul.” We have now added the affiliations of each group member and have indicated that Dr. McFarland is the group lead and added her email address. 

Reviewers' comments:

Reviewer #1: 

This is a retrospective single-center (US) cohort study over a 10-year period, which reports higher rate of congenital anomalies associated with exposure to integrase inhibitors in the first trimester compared to exposure to non-INSTI ART in the first trimester.

We thank Reviewer #1 for their careful review.

The main limitation of this article is that overall percentage of congenital anomalies is very high, even in non-INSTI-exposed infants. It would be important to provide baseline data for the US population and to make sure the categories used to report congenital anomalies in this paper meet standards for this kind of report.

We have added the background rate of birth defects in the US population that is reported by the MACDP to our discussion. In addition, we expanded our discussion on the limitations of registry/surveillance data (such as MACDP) and emphasized the reasons why we feel that the proportion of anomalies we identified in our cohort is accurate (lines 310-321): “We noted high overall rates of congenital anomalies in our cohort: 27.5% in INSTI-exposed and 11.9% in non-INSTI-exposed infants. In comparison, the MACDP estimates that major birth defects occur in approximately 3% of US births. Previous studies of HIV- and/or ART-exposed infants have documented rates of congenital anomalies ranging from 5.5 to 14.8%, suggesting that birth defects may occur more frequently in this population. One pharmacokinetics study of DTG in pregnant women diagnosed anomalies in 24% of exposed infants, similar to the proportion we identified. Importantly, surveillance programs that depend on registry data are subject to bias from under-reporting/differential reporting, under-ascertainment/differential ascertainment of birth defects, and loss to follow up. For example, the MACDP only captures defects identified upon surface examination at the time of delivery. We suspect that our clinic’s practice of following HIV-exposed infants with frequent visits and careful examinations over the first 12-18 months of life allowed us to identify anomalies that may have been overlooked in other settings, including anomalies that are not apparent upon surface examination.” In regards to whether the categories used to report congenital anomalies meet the standards for this type of report, we would direct the reviewer to lines 324-327: “Seventy percent of the anomalies we identified are diagnoses that are tracked by the MACDP, and diagnoses that are not tracked by the MACDP have been included in other reports of the association between congenital anomalies and INSTI exposure.”

Conclusion from this publication should be very careful as they are very different from previous reports. For instance, in a previous publication cited by this paper (Money 2019), there were 80 cases with dolutegravir exposure in the first trimester (including 69 cases with dolutegravir at conception) with four cases of non-chromosomal congenital anomalies, giving a rate of 5.0% (95% CI 1.4–12.3%) – this is highly different from this work.

As we described in our introduction (lines 71-74) and discussion (lines 315-317), analyses such as the one published by Money et al. that depend upon voluntary reporting are subject to limitations. We would point this reviewer to our previous response, in which we describe other publications that report much higher proportions of anomalies in other HIV and/or ART-exposed infants. 

Comparison of the group with exposure to integrase inhibitors in the first trimester to the group of exposure to non-INSTI ART in the first trimester is already done, but comparison to the group of no exposure to ART in the first trimester (n=90) should be added, while comparison to group never exposed to an INSTI should be removed.

As requested, we removed the analysis in which infants never exposed to an INSTI were used as the comparison group. However, given the current global recommendation to initiate treatment immediately in all people diagnosed with HIV, we do not feel that a comparison of outcomes among women who received INSTI versus no ART in the first trimester would provide any clinically applicable data. It seems more relevant for the reader to be able to compare the risk of adverse outcomes in pregnancies treated with INSTI versus the other ART options that would be used in their place. In addition, given that prior publications have identified a higher rate of anomalies in infants exposed to ART in utero versus the general population (see our references 30-32), we fear that using “no ART” as our comparison group would unfairly bias our analysis toward a perception of even higher rates of anomalies in INSTI-exposed infants. 

Further comments:

-Congenital anomalies should be categorized using the ICD https://icd.who.int/browse10/2016/en#/XVII.

As requested, we have re-categorized the anomalies according to ICD-10 codes, and updated our description of the anomalies in the text (lines 194-198), Table 2, and Figure 3. We would like to point out that separating “genital” from “urinary” anomalies resulted in a total of 50 (rather than 49) anomalies occuring among 36 infants. 

-Still birth should not be excluded from the analyses regarding congenital anomalies as autopsy data are available.

Our cohort included 2 stillbirths, one of whom had multiple dysmorphic features identified at autopsy, as we described in the manuscript. The mother of that stillbirth received a dolutegravir-based regimen beginning in the 2nd trimester. Because no ART was administered in the 1st trimester, it is not possible to include that stillbirth in our primary analysis (odds of an anomaly after 1st trimester INSTI exposure vs. 1st trimester non-INSTI ART exposure). The other stillbirth did not have an autopsy performed, only a surface examination, so anomalies cannot be ruled out with 100% certainty. For these reasons, we did not include the stillbirths in our analysis. 

-details the type of ART used in the non-INSTI group (what kind of a PI or NNRTI?). 

Table 2 lists the specific drugs received by the mothers of each infant who was diagnosed with a congenital anomaly (for both INSTI- and non-INSTI-based ART). 

-How was defined race/ethnicity in children and mothers ?

We added the following to our methods section (line 104-105): “Race and ethnicity were categorized according to maternal self-report.”

-Were the 36 infants with congenital anomalies tested for common causes of such anomalies, such as congenital CMV infection and genetic conditions? Of note, I am not sure that the trisomy 21 case should be excluded – either exclude all genetic conditions or none.

We added the following sentence to clarify our management of genetic conditions (lines 129-131): “Genetic screening to identify an etiology of congenital anomalies was performed at the discretion of the treating physicians; infants with a confirmed genetic condition were excluded from analyses of congenital anomalies.” The infant we identified with Trisomy 21 was the only infant with a genetic condition that was identified in this cohort. 

Regarding CMV, we added the following clarifying statements to the methods and results sections (lines 131-133; 200-201): “Infants whose mothers were seropositive or had unknown serostatus for cytomegalovirus (CMV) or Toxoplasma gondii were routinely screened for congenital CMV or T. gondii infection.” “None of the infants with congenital anomalies were diagnosed with congenital CMV or T. gondii infection.”

- Laboratory abnormalities: what is the protocol in place to follow lab toxicity in pregnant women living with HIV? How frequently were these lab tests performed in pregnancy in each group?

We added the following clarifying sentence (lines 112-113): “Maternal hematologic and complete metabolic panels were drawn at first visit during pregnancy, within 1 month of initiating or changing ART, and every 1-2 months throughout the duration of pregnancy.”

-Preeclampsia results: several other confounding factors should be addressed (BMI, parity, smoking…) – not sure you have enough cases for this.

Unfortunately, our ability to perform multivariable analyses of the preeclampsia outcome were limited both by small numbers and by our inability to collect data on all of the potential variables that might have been included, as we could only include data that were available in the electronic medical record. This is described in our limitations section (lines 367-372): “Our ability to perform multivariable analyses of the preeclampsia outcome was limited by the fact that we were unable to collect data on maternal body mass index, tobacco or alcohol use, or medications other than ART. However, few women in our cohort had other risk factors associated with preeclampsia, such as antiphospholipid antibody syndrome (n=0), pregestational diabetes (n=2), or assisted reproductive technology (n=0).”

-Table 1: Add gestational age at ART initiation (preconception versus first trimester versus 2 and 3rd trimester); Viral load at baseline and at delivery should be reported in caterogies (undetectable versus detectable)

The gestational age at start of maternal ART was added to Table 1. 

The cutoff for “undetectable” viral load varied between institutions where the laboratory test was run and across the years of the study. We used viral load as a continuous variable rather than a categorical variable to eliminate bias that could arise from differences in this cutoff. We added the following to the statistical methods section to explain how we handled samples with viral loads below the lower limit of detection (lines 140-144): “The lower limit of detection of HIV RNA PCR varied from <50 copies/mL to <20 copies/mL depending upon the year and the institution performing the laboratory test. When calculating median viral load, we used half of the lower limit of detection of HIV RNA PCR for samples with results below the lower limit of detection (i.e., 10 copies/mL for a sample reported as <20 copies/mL).”

-Table 2: Each infant should be represented only once, with a new category ‘Multisystem anomalies’ for those who have several anomalies; ; change also this in the text and in the abstract.

We feel that it is important to describe each specific diagnosis according to the body system that it affected. Therefore, we did not consolidate infants with more than one anomaly on a single row and describe them as having “multisystem anomalies,” as this would remove important details. Rather, we added a new column for “multisystem anomalies” and indicated the anomalies occurring in infants who had more than one anomaly type. The text and abstract are careful to point out that 50 anomalies occurred among 36 infants, and all of our statistical analyses of the association between anomalies and ART count each infant only once, as described in our statistical methods (lines 145-146): “We evaluated the association between DTG or any INSTI exposure in the 1st trimester or the 2nd/3rd trimester and the presence of at least one congenital anomaly (yes/no).”

-Table 3: Remove Polyhydramnnios, Oligohydramnios Placenta previa - Placental abruption (not relevant)

These outcomes have been removed from Table 3. 

Reviewer #2: 

Thank you for the opportunity to review this manuscript which is well written.

The manuscript describes associations between timing of INSTI-and non-INSTI ART on congenital anomalies in infants as well as other pregnancy outcomes, finding increased rates of congenital anomalies in infants born to women receiving INSTI-based ART pre-conception and in the first trimester, as well as increased pre-eclampsia.

We thank Reviewer #2 for their thoughtful review. 

Some minor revisions suggested:

Introduction:

-Line 45 suggest adding maternal health benefit to maternal ART (not just PMTCT)

The opening statement of our introduction has been revised to (lines 46-47): “Antiretroviral therapy (ART) is an invaluable tool for both preventing perinatal transmission of HIV and improving maternal health, but concerns exist regarding the maternal and fetal safety of these medications.”

-Line 61-63 suggest explaining clearly the issue with " lack of denominators" upfront, since this comes up a number of times in the manuscript but may not be understood by all and is unclear as written here. This becomes clearer in the sentence lines 70 onwards, but suggest that this explanation is included upfront.

We have revised the sentence in question as follows (lines 62-65): “Post-marketing surveillance data identified musculoskeletal, cardiac, and neurologic problems in infants exposed to DTG, but these reports are difficult to interpret given that the total number of infants exposed to these medications is unknown, so the proportion of infants with defects cannot be calculated.” 

- Suggest adding some information on background congenital anomaly rate in the community where the study took place

We added the background anomaly rate reported in the US to our discussion, and added an explanation of why we believe our anomaly rate was higher (lines 310-314): “We noted high overall rates of congenital anomalies in our cohort: 27.5% in INSTI-exposed and 11.9% in non-INSTI-exposed infants. In comparison, the MACDP estimates that major birth defects occur in approximately 3% of US births. Previous studies of HIV- and/or ART-exposed infants have documented rates of congenital anomalies ranging from 5.5 to 14.8%, suggesting that birth defects may occur more frequently in this population.”

Methods:

- Please include a brief standard of care explanation, with inclusion of whether women who are periconception receive folate supplementation in the periconception/1st TM through pregnancy? Also, whether there is grain fortification with folate? This could be a potential factor related to no NTD being seen?

We added the following sentence to our discussion (lines 366-367): “We did not collect data on maternal folate intake, but women in the CHIP program are encouraged to take prenatal vitamins, and grains in the U.S. are supplemented with folate.”

- Line 116-117 is not clear to me, specifically ".....starting a new medication or within 3 days of stopping a new medication". Should this be resolves within 3 days of stopping? That seems too short though? Please clarify

This sentence has been revised to (lines 121-123): “Laboratory abnormalities were considered potentially related to ARV exposure if they were identified for the first time at least 3 days after starting a new medication, at any time thereafter while taking the medication, or within 3 days after stopping the medication.”

- Obesity is an important factor related to congenital anomalies, particularly cardiac anomalies. Could the BMI not have been calculated using available weight and height, understanding that pre-conception weight may not have been available, but there is data from the first TM in many women.

Unfortunately, we were unable to capture height and weight data from enough women (at any point in pregnancy) to include BMI as a co-variable in our analyses. 

Table 2:

- This table is quite long-consider highlighting 1st TM exposures as well, rather than just pre-conception

We understand that Table 2 is quite long, but we felt it was important to describe each specific diagnosis according to the body system that it affected. We are happy to move Table 2 to the supplementary materials if the editor thinks it is more appropriate there. The final column of Table 2 describes the gestational age at which maternal ART was started for ALL women – not just those who started ART preconception. To better “highlight” 1st trimester exposures, we added “(1st trimester)” beside all of the gestational ages of ART initiation that were ≤13 weeks. 

Discussion

- Line 294-296: point well taken, but perhaps also specify that many of the pregnancy registries collect surface examination data only at delivery/first few weeks and have the advantage of larger numbers, but do not investigate for internal anomalies that are not visible.

We thank the reviewer for this excellent point, which has been added to the discussion (lines 315-321): “Importantly, surveillance programs that depend on registry data are subject to bias from under-reporting/differential reporting, under-ascertainment/differential ascertainment of birth defects, and loss to follow up. For example, the MACDP only captures defects identified upon surface examination at the time of delivery. We suspect that our clinic’s practice of following HIV-exposed infants with frequent visits and careful examinations over the first 12-18 months of life allowed us to identify anomalies that may have been overlooked in other settings, including anomalies that are not apparent upon surface examination.”

- Obesity data not available is mentioned as a limitation, although please see previous comments regarding calculation which should be done if possible.

Unfortunately this calculation is not possible, as we described above. 

Reviewer #3: 

The paper is interesting but as the authors point out in the study limitations the cohort has many limitations, not the least of which is the failure to report alcohol and smoking for example

We thank Reviewer #3 for their thoughtful review. 

We agree that it is unfortunate that we do not have access to data on alcohol or tobacco use. This is reported as a limitation (lines 367-370): “Our ability to perform multivariable analyses of the preeclampsia outcome was limited by the fact that we were unable to collect data on maternal body mass index, tobacco or alcohol use, or medications other than ART.”

To make it definitively acceptable the authors could check only major congenital defects, in particular major anomalies were defined as those that required surgical or medical intervention or had serious cosmetic significance.

The original version of our manuscript included a sensitivity analysis that restricted to defects that are recognized by the MACDP (see lines 217-219, 235-238). This analysis found a trend toward increased anomalies in infants exposed to INSTIs, although the association was no longer significant, likely as a result of smaller numbers in the analysis. We suspect that we would find a similar non-significant trend if we performed a sensitivity analysis restricting to major anomalies, considering that even fewer of the anomalies we identified were considered major (54%) than were diagnoses tracked by the MACDP (70%). In addition, the benefit of a sensitivity analysis restricting to major anomalies would be to get a sense of the healthcare burden of having to address these anomalies; however, it would not help clarify the underlying question of whether particular drugs cause anomalies in the first place. For these reasons, we decided not to pursue this additional sensitivity analysis. 

Also with regard to the diagnosis of preeclampsia, risk factors are missing: antiphospholipid antibody syndrome, Chronic hypertension, Pregestational diabetes and body mass index (BMI) >30 and assisted reproductive technology. (Bartsch E, Medcalf K E, Park A L, Ray J G. Clinical risk factors for pre-eclampsia determined in early pregnancy: systematic review and meta-analysis of large cohort studies BMJ 2016; 353 :i1753 doi:10.1136/bmj.i1753).

We have added the following information about why chronic hypertension was not included in the multivariable analysis (lines 257-260): “Of note, four women had pre-existing chronic hypertension (two women received INSTI-containing ART and each had one pregnancy, two women received non-INSTI ART and had a total of 4 pregnancies); these numbers were too small to include hypertension as a co-variable in the multivariable model.” We agree that it is unfortunate that we do not have access to data on body mass index. This is reported as a limitation (lines 367-370): “Our ability to perform multivariable analyses of the preeclampsia outcome was limited by the fact that we were unable to collect data on maternal body mass index, tobacco or alcohol use, or medications other than ART.” We have added a sentence addressing the other risk factors and cited the article that was recommended by the reviewer (lines 370-372): “However, few women in our cohort had other risk factors associated with preeclampsia, such as antiphospholipid antibody syndrome (n=0), pregestational diabetes (n=2), or assisted reproductive technology (n=0).”

Finally, if the authors are able to complete these requirements, it would also be helpful not to overemphasize the role of INSTIs in the title, but simply report the results of their single-center cohort with many limitations.

We changed our title to “Congenital Malformations and Preeclampsia Associated with Integrase Inhibitor Use in Pregnancy: A Single-Center Analysis” to note that the association we found was in a single center cohort. 

We appreciate your consideration to the submission of this manuscript and we look forward to hearing from you.

---

## [Editor Report · Decision Letter 1]

30 May 2023

Congenital Malformations and Preeclampsia Associated with Integrase Inhibitor Use in Pregnancy: A Single-Center Analysis

PONE-D-22-27763R1

Dear Dr. Smith,

We’re pleased to inform you that your manuscript has been judged scientifically suitable for publication and will be formally accepted for publication once it meets all outstanding technical requirements.

Kind regards,

Carlo Torti

Academic Editor

PLOS ONE

Additional Editor Comments (optional):

This paper is suitable for publication.
---

## [Editor Report · Acceptance letter]

4 Jun 2023

PONE-D-22-27763R1 

Congenital Malformations and Preeclampsia Associated with Integrase Inhibitor Use in Pregnancy: A Single-Center Analysis 

Dear Dr. Smith:

I'm pleased to inform you that your manuscript has been deemed suitable for publication in PLOS ONE. Congratulations! Your manuscript is now with our production department. 

Kind regards, 

on behalf of

Dr. Carlo Torti 

Academic Editor

PLOS ONE